# Work-Related Traumatic Stress Response in Nurses Employed in COVID-19 Settings

**DOI:** 10.3390/ijerph191711049

**Published:** 2022-09-03

**Authors:** Maria Karanikola, Meropi Mpouzika, Elizabeth Papathanassoglou, Katerina Kaikoushi, Anna Hatzioannou, Ioannis Leontiou, Chris Livadiotis, Nicos Christophorou, Andreas Chatzittofis

**Affiliations:** 1Department of Nursing, School of Health Sciences, Cyprus University of Technology, Limassol 3041, Cyprus; 2Faculty of Nursing, University of Alberta, Edmonton, AB T6G 1C9, Canada; 3Cyprus Community Mental Health Services, Famagusta 5566, Cyprus; 4Limassol General Hospital, Limassol 3041, Cyprus; 5Medical School, University of Cyprus, Nicosia 1065, Cyprus; 6Department of Clinical Sciences/Psychiatry, Umeå University, 901 85 Umeå, Sweden

**Keywords:** COVID-19 settings, emotional exhaustion, job satisfaction, nurses, organizational support, traumatic stress symptoms, secondary trauma

## Abstract

Nurses may be at a higher risk of experiencing work-related traumatic stress response during the COVID-19 pandemic compared to other clinicians. This study aimed to investigate the correlations between work-related trauma symptoms and demographic factors, psychosocial hazards and stress response in a census sample of nurses working in COVID-19 settings in Cyprus. In this nationwide descriptive and cross-sectional study, data were collected between April and May 2020 using a questionnaire that included sociodemographic, educational and employment and work-related variables, as well as a modified version of the Secondary Traumatic Stress Scale (STSS) for the assessment of work-related trauma symptoms during the pandemic. Overall, 233 nurses participated (with a response rate of 61.3%) and 25.7% of them reported clinical work-related trauma symptoms (STSS-M > 55; actual scale range: 17–85). The mean value for emotional exhaustion was 7.3 (SD: 2.29; visual scale range: 1–10), while the value for distress that was caused by being avoided due to work in COVID-19 units was 6.98 (SD: 2.69; visual scale range: 1–10). Positive associations were noted between trauma symptoms and both emotional exhaustion and distress from being avoided by others due to work in a COVID-19 setting and a negative association was also found between trauma symptoms and satisfaction from organizational support variables (all *p* < 0.002). Working in COVID-19 settings during the pandemic is a stressful experience that has been linked to psychologically traumatic symptoms Thus, supportive measures are proposed for healthcare personnel, even in countries with low COVID-19 burden.

## 1. Introduction

The COVID-19 pandemic has been a challenge to healthcare systems and healthcare workers, including nurses, as it has increased work-related psychological hazards and has affected the physical and mental health of clinicians [1,2,3,4,5]. Work-related psychosocial hazards are defined as factors mainly related to the management or design of work environments that increase the risk of work-related stress response, such as high job demands or inadequate managerial support, and may cause physical or psychological harm to employees [6].

Work-related psychosocial hazards may adversely affect the health and well-being of clinicians by causing stress response; however, although psychosocial hazards may be present in work environments, they may not be experienced as threats and thus, may not cause any stress response [6,7]. Therefore, the distinction between psychosocial hazards and those factors which may actually cause work-related stress response is important. Specifically, the causes of work-related stress response, which are otherwise called work-related stressors, may be specific to clinicians (e.g., mental and psychological difficulties, low emotional intelligence or low resilience status) or to workplace features (e.g., shift work, dysfunctional professional relationships or demanding work settings, such as intensive care units (ICUs) or COVID-19 departments) [6,8,9,10,11]. Thus, it is imperative to clearly distinguish between work-related psychosocial hazards, work-related stressors and their health implications on those who are employed in COVID-19 settings, including stress response.

### 1.1. Work-Related Psychosocial Hazards and Work-Related Stress Response in Healthcare Professionals Working in COVID-19 Settings 

Clinicians who are employed in COVID-19 settings are exposed to increased workload; care rationing [12,13]; the need to cope with inadequate resources and funds; and to a rapidly changing healthcare organizational status in terms of policies and procedures, both administrative and therapeutic [14]. Additionally, beyond the fear of suffering from COVID-19 themselves, concerns regarding transmitting COVID-19 to family members and the social isolation from supportive networks due to protective measures are also prominent among clinicians [2,12,13].

All the above factors are associated with decreased work satisfaction and the intention to quit the job [15,16]; at the same time, they may trigger physical, psychological and mental disturbances in healthcare professionals, such as restlessness, worry, insomnia, burnout, moral distress, compassion fatigue, anxiety and depressive symptoms [17,18,19,20,21].

### 1.2. Work-Related Traumatic Stress Response in Healthcare Professionals Working in COVID-19 Settings

Work in COVID-19 settings may involve exposure to traumatic events [13]. Work-related traumatic events are incidents that may cause intense fear, or severe distress among employees and include the threat of harm or actual harm, and/or exposure to abuse [22]. Intense fear and severe distress from direct or indirect exposure to traumatic events may lead to psychological or physical injury, which is mostly known as work-related traumatic stress response [21]. Indeed, clinicians who are employed in COVID-19 settings have close encounter with this communicable and life-threatening disease; they are constantly witnessing serious physical injuries in their patients while experiencing the fear of contracting the illness themselves. Thus, caring for patients with COVID-19 is deemed as a potentially work-related traumatic experience for clinicians [7,8]. Nevertheless, researchers have suggested that the COVID-19 pandemic may also be interpreted as a collective traumatic event, which may trigger symptoms that are related to post-traumatic stress disorder (PTSD) [23]. Moreover, data have shown close links between the intrusion symptoms of PTSD, mental health status and the fear of COVID-19, which is further mediated by hyperarousal and avoidance symptoms [23].

Post-traumatic stress disorder is defined as a clinical condition that occurs in individuals who have been exposed to severely traumatic events, including work-related events [22]. According to the fifth edition of the Diagnostic and Statistical Manual of Mental Disorders (DSM-5), the first criterion for the diagnosis of PTSD is exposure to actual or threatened death, serious injury or sexual violence in one of the following ways: (a) direct experience of a traumatic event; (b) witnessing a traumatic event as it occurs to others; (c) learning that a traumatic (accidental or violent) event has occurred to a close family member/friend; (d) the experience of extreme or repeated exposure to the aversive details of a traumatic event [22]. Based on this criterion, it is clear that PTSD encompasses both direct and indirect exposure to traumatic events, as well as the vicarious (secondary) trauma. 

Vicarious (secondary) trauma includes adverse impacts from indirect exposure to potentially traumatic events in the workplace, such as caring for service users who are coping with traumatic events, reviewing distressing information or witnessing a fatality, as well as knowing about or witnessing the suffering of others, all of which are relevant to healthcare employees during the pandemic [24]. Since nurses who are employed in COVID-19 settings experience both direct and indirect exposure to serious threats, “work-related traumatic stress response” has been proposed as a more integrative and inclusive term to reflect their overall exposure to adverse work conditions during the pandemic [21,24]. 

Several tools exist for the assessment of the severity of PTSD symptoms, such as the Impact of Events Scale or the Post-Traumatic Symptom Scale, which have been used in research samples of healthcare workers who were assessed for vicarious (secondary) trauma [21]. 

Although different factors, such as personal characteristics and work-related factors, have been suggested to play a role in developing symptoms of mental distress among clinicians who have been exposed to traumatic events [25,26], only a few studies have investigated the link between traumatic stress response and work-related psychosocial hazards and/or work-related stress response in clinicians who are employed in COVID-19 settings [27,28,29]. Additionally, previous studies on this group of clinicians have mainly focused on depressive and stress symptoms and have neglected organizational factors and work satisfaction [30]. However, there is evidence of lower self-perceived organizational support among less experienced, female and non-physician healthcare workers, which is also inversely associated with stress response and depressive and traumatic symptoms [31,32]. Furthermore, there have not been many studies focusing specifically on traumatic stress response in nurses working in COVID-19 settings. Previous studies have mostly enrolled mixed samples of healthcare workers who were employed in ICUs or the emergency medical services [33,34]. Yet, data have revealed that staff nurses self-report higher levels of depressive, anxiety and burnout symptoms compared to other healthcare professionals, which underlines the need for additional research on this group regarding work-related traumatic stress response symptoms [35,36].

Nevertheless, existing data calls for attention to be paid to the social, relational and environmental context of clinicians to better understand their distress and their risk of developing mental health problems during and after the pandemic [37]. Similarly, since the pandemic has forced healthcare systems around the world to reform their mental health services to varying degrees in order to address emerging needs [38], additional data is needed to support these adaptations. 

### 1.3. The Present Study: Aim and Objectives

Τhe aim of the present study was to assess the prevalence and intensity of work-related traumatic stress response in Greek Cypriot nurses who were employed in COVID-19 settings. Our secondary objectives included the following: (i) an exploration of the associations between the presence and intensity of traumatic stress response and (a) sociodemographic variables, (b) work satisfaction and (c) self-assessed work-related psychosocial hazards (e.g., satisfaction from personal protective equipment, information provided and care); (ii) an exploration of self-assessed work-related stress response (e.g., emotional exhaustion and distress caused by being avoided due to work in a COVID-19 setting).

## 2. Materials and Methods

### 2.1. Design

A descriptive and cross-sectional design was applied in this study. The Strengthening the Reporting of Observational Studies in Epidemiology (STROBE) guidelines were followed [39].

### 2.2. Study Settings and Context

The study population comprised nurses working in COVID-19 settings in public healthcare services in the Republic of Cyprus (RC). The Nicosia General Hospital (NGH) is the largest public tertiary hospital in the RC and is based in Nicosia. Its emergency department (ED) (which employs 12 physicians and 72 nurses) provides services to over 140 patients per day and, in parallel, it was the referral center for those reporting COVID-19 symptoms during the first wave of the pandemic. Following assessment in the ED, patients who produced positive SARS-CoV-2 tests were either admitted to an ICU (one is based in the NGH and one is in the Limassol General Hospital) or they were transferred to (a) one of the two open COVID-19 units in the NGH or (b) the COVID-19 referral hospital in Famagusta, according to the severity of their symptoms. The Famagusta referral hospital ran one step-down unit (i.e., an open unit for advanced care) and four open wards. Nurses from all of the above settings were invited and participated in the present study (i.e., the two ICUs, five open units, one step-down ICU and one ED).

### 2.3. Sample Size and Sampling

A census sampling approach was applied. Based on power analysis, the sample size had to be 104 nurses for moderate correlation effect (0.3–0.4) to be demonstrated between work-related traumatic stress response and work, employment and demographic variables (with an 80% statistical power and a statistical significance level of 0.05 in the multivariate analysis). By taking previous data on the response rate of Greek Cypriot nurses into consideration [40], a total of 380 questionnaires were distributed.

### 2.4. Participants

Our inclusion criteria included (a) employment in a COVID-19 setting (EDs, ICUs/step-down ICUs or open COVID-19 units), (b) the ability to read and write in Greek and (c) the provision of written informed consent. There were no exclusion criteria.

### 2.5. Variables and Measurements

Sociodemographic (age, gender, marital status and number of children), education (education level) and employment (type of work setting/unit, work province, years of work experience and ranking) variables were assessed by using closed-ended questions. The following work variables were approximately reported by responders: the number of patients treated per work setting; the number of deaths per day in the last month in the work setting from any cause; the number of night shifts per month; the number of days in quarantine after producing a positive SARS-CoV-2 test/self-quarantine due to exposure to SARS-CoV-2 (close contact). 

The degrees of (a) emotional exhaustion, (b) work satisfaction, (c) satisfaction from care provided in the last month, (d) satisfaction from information provided about COVID-19, (e) satisfaction from the provision of personal protective equipment by the work organization and (f) distress caused by being avoided due to work in COVID-19 healthcare settings were assessed by using visual analogue scales (VASs). The VAS scores ranged from 1 to 10, with higher values indicating increased satisfaction/distress. The VAS tools that were used for the measurement of the degree of emotional exhaustion, work satisfaction and satisfaction from provided care have been validated in previous studies in Greek-Cypriot nurses [40]. The VAS tools that were used for the measurement of the degree of satisfaction from information provided about COVID-19, satisfaction from personal protective equipment and distress caused by being avoided due to work in COVID-19 healthcare settings were developed for the purpose of the present study, according to the relevant literature [2,7,12,13,17], and were validated by a group of experts (content validity).

The modified 17-item version of the Secondary Traumatic Stress Scale (STSS-M) was used for the assessment of work-related traumatic stress response. The items that were included in the original STSS were modified accordingly to be suitable to experiences of psychological trauma that were related to work in COVID-19 settings. Examples of the modified statements included “Reminders of my work with clients in the COVID-19 unit upset me” and “My heart started pounding when I thought about my work with clients in the COVID-19 unit”. The items were rated on a 5-point Likert scale, from 1 (never) to 5 (very often). The participants were asked to report the frequency at which they experienced the situations that were described in each item during the previous two weeks. The total STSS-M score ranges from 17 to 85, with higher scores indicating more intense self-reported trauma symptoms (TS). The 17 items of the instrument could be grouped into avoidance, arousal and intrusion symptoms, which reflected the three subscales of the STSS-M (not presented in the present analysis). The total STSS-M scores were interpreted by comparing the scores of the study participants to the normative scores. Specifically, the responses of the participants were classified into categories based on the percentiles: total STSS-M scores that were at or below the 50th percentile (i.e., less than 46) were interpreted as being indicative of no or low-intensity trauma symptoms; scores between the 51st and 75th percentiles (i.e., 46 to 55) were interpreted as mild-intensity trauma symptoms; scores between the 76th and 90th percentiles (i.e., 56 to 62) were interpreted as moderate-intensity trauma symptoms; scores between the 91st and 95th percentiles (i.e., 63 to 67) were interpreted as high-intensity trauma symptoms; scores that were above the 95th percentile (i.e., above 67) were interpreted as severe-intensity trauma symptoms. Furthermore, total STSS-M scores that were at or above the cutoff value of 56 (i.e., the lowest threshold of the moderate intensity range) were indicative of clinically relevant traumatic stress symptoms that were caused by exposure to COVID-19-related work conditions. 

The internal consistency reliability coefficient (Cronbach’s alpha) of the original STSS has been reported to be as high as 0.93, and 0.9 for relevant modified versions [21,41]. Previous data also supported the construct validity of the original tool [41]. The Cronbach’s alpha for the STSS-M that was applied in this study was 0.91.

### 2.6. Data Collection and Instrument

Data collection took place in April and May 2020 via a self-administered paper questionnaire that was distributed by the researchers to participants in the study work settings. The response rate was 61.3%. 

### 2.7. Ethical Issues

Written informed consent was provided by all participants. Anonymity and the voluntary nature of participation in the study were also assured. Each questionnaire package was provided in an open and opaque envelope with no identifying characteristics. The study was conducted according to the guidelines of the Declaration of Helsinki. Permission to carry out the study was obtained from the National Committee of Bioethics of Cyprus [EEBK/EP/2019/27].

### 2.8. Data Analysis

Frequencies were assessed for categorical variables and mean values (M) and standard deviations (SD) were assessed for continuous variables. The age and years of work experience were transformed from continuous into categorical variables. The following age groups were formed: up to 35 years; 35–45 years; more than 45 years. Work experience was grouped as follows: less than 5 years; 5–10 years; more than 10 years. The continuous variables were checked for normality and the *t*-test and ANOVA parametric measures were applied for comparisons between the groups. For significant differences between multiple groups, post-hoc analyses were carried out by using the Scheffe test. The chi-squared test was used for comparisons between groups regarding the categorical variables. Pearson’s r was assessed to determine the correlations between the numerical variables. Aiming to model the predictors of work-related traumatic stress response (total STSS-M score), a multivariate analysis was applied by using stepwise logistic regression. Specifically, the total STSS-M score was transformed into a categorical dichotomous variable (dependent variable) as follows: no clinical TS (17–55 STSS-M score); clinical TS (56-85 STSS-M score). The following were also included as independent variables: (a) sociodemographic, educational, work-related and employment variables; (b) VAS variables, as described in the Data Analysis section. The level of statistical significance was set at < 0.05. The IBM SPSS (version 25.0, IBM, Chicago, IL, USA) statistical package was used for the data analysis. 

## 3. Results

### 3.1. Sociodemographic, Work and Employment Characteristics of Participants

The sample consisted of 233 nurses: 86 men (36.9%) and 147 (63.1%) women. The majority were up to 35 years old (58.4%) and 57% (n = 133) had more than 10 years of work experience. These data are presented in Table 1. 

Approximately 75.1% of the participants reported that they treated fewer than 10 patients per shift and 89.7% reported fewer than five daily deaths from any cause during the previous month. A total of 129 participants (55.4%) worked more than five night shifts per month during the same period. Approximately 47% were employed in an ICU, 22% were employed in an ED, 25% were employed in an open COVID-19 unit and 6% were employed in a step-down ICU.

The mean number of days per month that were spent in quarantine after a positive SARS-CoV-2 test was 10.04 (SD: 13.28; range: 0–40), while the mean number of days that were spent in self-quarantine due to exposure to SARS-CoV-2 (close contact) was 3.10 (SD: 6.56; range: 0–31). The sociodemographic and employment characteristics of the study participants are presented in Table 1.

### 3.2. Work-Related Psychosocial Hazards and Work-Related Stress Response

The average values for the variables “Degree of emotional exhaustion” (mean: 7.33; SD: 2.29) and “Degree of distress experienced from being avoided due to work in a COVID-19 healthcare setting during the pandemic” (mean: 6.98; SD: 2.69) were above moderate (scale range 1–10), as was the mean value for “Degree of satisfaction from provided care” (mean: 6.87; SD: 2.02). All of the VAS scores are presented in Table 2. Additional data on the differences in the mean scores of satisfaction and distress that was experienced from work-related factors and the groups of sociodemographic, educational and employment variables are presented in Appendix A in the Appendix A.

### 3.3. Work-Related Traumatic Stress Response (Total STSS-M Scores) 

The mean total STSS-M score was 45.38 (SD: 12.97; range: 17–80). According to the STSS-M scoring classification, 23.6% (n = 55) had no or low-intensity TS, 25.8% (n = 60) had mild-intensity TS and 24.9% (n = 58) had moderate-intensity TS (i.e., STSS-M scores < 56), while 15.0% (n = 35) had high-intensity TS and 10.7% (n = 25) had severe-intensity TS (i.e., STSS-M scores > 55) (Table 3). 

Female participants reported more intense TS compared to male participants (mean (SD): 47.01 (12.53) vs. 42.61 (13.30); *p* = 0.014)) and the age group up to 35 years (61.5%) reported more intense TS compared to the other age groups (*p* = 0. 016). Participants who reported more than five daily deaths reported more intense TS compared to those who reported fewer than five daily deaths (mean (SD): 50.55 (13.83) vs. 44.90 (12.83); *p* = 0.05)). There were no associations between these variables and work setting. Additional data on the differences in the severity of work-related traumatic stress response (total STSS-M scores) according to demographic characteristics (age, gender, etc.) are presented in Appendix A.

### 3.4. Associations between STSS-M Scores and Work-Related Satisfaction and Distress 

The correlations between the total STSS-M scores and the VAS scores for work-related psychosocial hazards and work-related stress response are reported in Table 4. Emotional exhaustion and distress experienced from being avoided due to work in COVID-19 units were moderately and positively correlated with the intensity of traumatic stress response symptoms (*p* < 0.001). STSS-M score was inversely correlated with the degree of satisfaction from provided care in the last month, the degree of satisfaction from information provided about COVID-19 by the administrative managers and the degree of satisfaction from the provided personal protective equipment against COVID-19 (*p* < 0.002). No correlations were found between the total STSS-M score and work/employment variables (Table 4.).

### 3.5. Multivariable Analysis of Predictors of Work-Related Traumatic Stress Response (STSS-M Score)

A stepwise logistic regression was applied, and the results are presented in Table 5. Being a staff nurse and experiencing higher emotional exhaustion, lower work satisfaction and higher distress from being avoided due to work in COVID-19 settings were all independent predictors of clinically relevant work-related traumatic stress symptoms (STSS-M score > 55) (Table 5).

## 4. Discussion

In this nationwide study, we investigated the prevalence and intensity of work-related traumatic stress response symptoms in a large sample of nurses who were employed in COVID-19 settings. We found that almost one out of four of these nurses self-reported severe trauma symptoms that were related to their work during the pandemic. We measured the work-related traumatic stress response symptoms using a tool that was specifically designed to assess both direct and indirect exposure to work-related traumatic events. Other studies have used tools that were not specific to work conditions. For instance, Bani Issa et al. (2021) [42] found that 36.2% of participants in a sample of 370 nurses self-reported clinically relevant trauma symptoms, as assessed by the Post-Traumatic Diagnostic Scale. Similarly, Crowe et al. (2021) [43] reported that 37.6% of nurses in their sample experienced severe traumatic stress symptoms, as assessed by the Impact of Event Scale–Revised. 

Our results were in accordance with those from previous research on work-related traumatic stress response symptoms in mixed samples of healthcare workers during the first wave of the pandemic [21,32]. Specifically, the review by d’Ettorre et al. (2021) [32] of 14 studies that analyzed the occurrence of traumatic stress response symptoms in healthcare workers revealed that a range between 2.1% and 73.4% of participants who experienced clinically relevant symptoms. Any variations in the reported occurrence of trauma symptoms in comparison to that in previous studies could still be adequately explained by (a) the different instruments that were applied for the measurement of work-related traumatic stress response symptoms, (b) the timing of data collection, (c) the study designs and (d) the work settings. Indeed, lower rates of work-related traumatic stress response symptoms in healthcare professionals have been noted at the very beginning of the pandemic and between outbreaks, as opposed to at the peaks of pandemic waves. The present study took place during the peak of the first wave of the pandemic and reported a rate that was similar to those in the majority of the studies that were reviewed by d’Ettore et al. (2021) [32]. 

We also found that staff rank, emotional exhaustion, low work satisfaction and distress from being avoided due to work in a COVID-19 setting were associated with clinically relevant work-related traumatic stress response symptoms. The perception of being avoided by others due to work in a COVID-19 setting could be deemed as an aspect of poor social support and could underline the social implications of the pandemic on nurses, which have also been addressed in other studies [44]. Overall, data have shown a strong relation between diminished social support and work-related traumatic stress response symptoms in healthcare workers during the first wave of the pandemic [32]. 

In terms of staff rank, our study revealed that participants with leadership roles were less vulnerable to work-related traumatic stress response symptoms than those who held a staff nurse position, which was contrary to the finding in a study by Inocian et al. (2021) [45]. One possible explanation for this could be that being a nurse with a leadership role allowed for participation in decision making and, subsequently, improved professional experience [46]. Furthermore, participants who declared increased professional satisfaction reported lower STSS-M scores, which was in line with previous findings [47].

The positive association between work-related traumatic stress response symptoms and emotional exhaustion that was reported by this study was in accordance with the finding of a study by Luceño-Moreno et al. (2020) [48]. The psychological burden of emotional exhaustion and adverse emotions, such as fear, distress and insecurity, have also been found in nurses who have been exposed to survivors of past viral epidemic conditions, as well as survivors of the present pandemic [44,49]. Dysfunctional thoughts that trigger tension and emotional distress in healthcare professionals include the following: “I am concerned that I may transmit the virus to my family, or to my colleagues”; “people avoid me or will avoid me in the future because they may be afraid that I will transmit COVID-19 to them”; “I am concerned that I do not have the appropriate knowledge and/or the appropriate means/equipment to protect myself from COVID-19” [44]. A loss of perspective, pessimism and spiritual distress have also been reported in clinicians, as well as experiences of moral distress that were relevant to care rationing [14]. Overall, these difficulties seem to compromise the ability of nurses to fulfil their roles at a professional level, as well as at personal, familial and social level [50]. 

The present study did not reveal any predictors of traumatic stress symptoms that were related to sociodemographic or employment variables, although previous reviews have shown that female staff nurses working in EDs and ICUs during the pandemic reported higher levels of depressive, anxiety and burnout symptoms compared to male healthcare professionals (nurses or otherwise) employed in non-ICU/non-ED settings [35,36]. Other studies have reported that traumatic stress response symptoms in healthcare workers during the pandemic were associated with heavy workloads, young age, female gender, a lack of training and diminished social support [32,48,51,52,53,54]. The present study also found that young female nurses who reported more than five daily deaths experienced more intense traumatic stress symptoms; however, these variables did not remain important predictors of clinically relevant traumatic stress response symptoms in the multivariate regression analysis. One possible explanation for this could be that the present study included additional covariates, such as work satisfaction and emotional exhaustion, which could have exerted stronger effects on the phenomenon of work-related trauma than personal characteristics. Moreover, this difference also supported the importance of organizational factors over individual factors, such as age or gender, in triggering work-related traumatic stress symptoms.

In this study we also reported that the degrees of satisfaction from provided care, information about COVID-19 and the supply of personal protective equipment were negatively related to work-related traumatic stress symptoms. Other studies have also underlined the link between organizational support measures and mental health in healthcare professionals. Providing protective equipment, together with supporting the needs of healthcare workers, have also been highlighted as crucial for engagement in work and the provision of optimal care to patients [55]. Moreover, organizational support for healthcare professionals has been positively associated with nurses’ adaptability during the COVID-19 pandemic and has emerged as a strong predictor of mental health symptoms, including post-traumatic stress symptoms, in frontline clinicians [56,57,58]. In particular, three organizational factors (i.e., “work support”, “personal support” and “risk support”) have been identified as being inversely related to anxiety, while “work support” and “personal support” have also been shown to predict higher life satisfaction among clinicians [59]. 

### 4.1. Future Research Directions

Although healthcare systems around the world have reformed their mental health services to varying degrees due to pressure from the pandemic, there is still a lot to be improved [38]. Our study revealed some data that could be used to develop interventions. Working in COVID-19 settings during the pandemic is a stressful experience and has been linked to psychologically traumatic symptoms; thus, it is imperative to develop national support and recovery strategies that are aimed at afflicted healthcare professionals [20]. One suggestion includes the education of frontline nurses regarding self-care approaches to enhance their resilience to traumatic symptoms via national strategic planning. This strategy is proposed to be designed through a participatory action research approach, and to include frontline nurses to its design. This inclusion is expected to enhance nurses’ engagement in the strategy, thereby allowing it to have a greater impact. In a recent narrative review, a number of personal approaches and self-coping strategies, such as active planning, behavioral disengagement and physical exercise, have been shown to be effective in controlling the psychological burden of healthcare employees during the pandemic [60]. Nevertheless, in an effort to diminish the physiological, psychological and mental burden of future outbreaks of infectious diseases, healthcare employees need to be prepared and educated on how to defend and promote their physiological, psychological and mental well-being [60].

Furthermore, based on the presented results and relevant literature, leadership approaches that could reduce the traumatic impact of work-related psychosocial hazards on frontline nurses include (a) empowerment to participate in decision making and access to information, (b) the provision of adequate personal protective equipment and (c) education about COVID-19 and relevant copying strategies and the development of a COVID-19 workplace protocol to promote safe and well-organized work environments [16,56]. The above approaches should be adopted by leaders at a collective level to empower healthcare personnel and support healthcare organizations [61]. 

Additionally, nurses who are working on the frontline during the pandemic should be closely monitored for the development of mental health problems. The early identification of traumatic stress response symptoms is important as these symptoms are related to burnout, poor quality of work life and the intention to quit the job [62]. In cases of identified traumatic stress response symptoms, professional mental health support is needed [63]. For instance, the Center for the Study of Traumatic Stress in Bethesda, Maryland, offers insightful and evidence-based psychological first aid principles [64].

Lastly, longitudinal studies are needed to explore the trajectory of work-related traumatic stress responses in nurses who are employed in work environments that have increased psychosocial hazards and a multitude of causes of work-related stress responses.

### 4.2. Strengths and Limitations

The strengths of the present study included the simultaneous assessment of work-related traumatic stress response symptoms during the pandemic, work-related psychosocial hazards and stress response in a large nationwide study that focused on nurses who were employed in COVID-19 healthcare settings in the Republic of Cyprus. Moreover, the sampling method that was applied supported the generalization of the presented results. There was also a large response rate, which minimized any possible selection bias. However, although the questionnaire was distributed to all nurses, selection bias could not be totally excluded. Additional limitations of this study included its cross-sectional design, which limited etiological interpretations, the self-reported assessment of trauma symptoms and possible impact of cultural factors, as the study was conducted in the multicultural Republic of Cyprus. Moreover, we were not aware of the characteristics of the nurses who eventually did not consent to participation; thus, our results need to be replicated in independent studies and settings. Additionally, although three of the VASs that were applied in this study were developed by the authors, they were based on the relevant literature, and were validated by a group of experts. Nevertheless, additional validation methods could have increased the internal validity of the study. Similarly, it is worth mentioning that the developer of the original STSS did not have any input during the development of our modified version.

Finally, additional confounders that could have contributed to the reported traumatic symptoms, such as psychiatric history or previous traumatic experiences, were not taken into consideration, nor were the baseline measurements of the mental health status of the participants prior to the onset of the pandemic. Nonetheless, the present study is among the few that have addressed work-related psychosocial hazards (e.g., organizational factors) and stress response in relation to work-related traumatic stress response symptoms specifically in nurses who were employed in COVID-19 settings.

## 5. Conclusions

The results of this study suggested that nurses who are employed in COVID-19 settings are vulnerable to developing work-related traumatic stress response symptoms. Supportive environments in the workplace need to be assured by administrators, even in countries with low COVID-19 burden, while empowerment interventions and relevant facilities need to be available for those who are most at risk, i.e., those in COVID-19 referral centers and those who have the closest contact with COVID-19 patients.

## Figures and Tables

**Table 1 ijerph-19-11049-t001:** The sociodemographic and employment characteristics of the sample (N = 233).

	*N*	*Percentage (%)*
**Gender**		
Male	86	36.9
Female	147	63.1
**Age**		
Up to 35 years	136	58.4
35–45 years	60	25.7
>45 years	37	15.9
**City of Employment**		
Nicosia	124	53.2
Limassol	65	27.9
Famagusta (COVID-19 Referral Hospital)	44	18.9
**Marital Status**		
Married	157	67.4
Unmarried	76	32.6
**Number of Children**		
No children	77	33.0
1–3 children	146	62.7
>3 children	10	4.3
**Education**		
No post-graduate education	172	73.8
Post-graduate education	61	26.2
**Total Work Experience in Nursing**		
<5 years	12	5.2
5–10 years	88	37.8
>10 years	133	57.0
**Ranking**		
Staff nurse	218	93.6
Head nurse/under head nurse	15	6.4
**Total Number of Patients Treated per Work Setting**		
<10 patients	175	75.1
10–20 patients	38	16.3
>20 patients	20	8.6
**Number of Deaths per Day in the Previous Month from Any Cause**		
<5	209	89.7
>5	24	10.3
**Number of Night Shifts per Month**		
<5	104	44.6
>5	129	55.4

**Table 2 ijerph-19-11049-t002:** The satisfaction and distress that were experienced from work-related factors (Ν = 233).

*Visual Analogue Scales*	*Median*	*Mean*	*St. Deviation*
** *Degree of Experienced Satisfaction* **			
Degree of work satisfaction	6.00	6.21	2.16
Degree of satisfaction from provided care in the last month	7.00	6.87	2.02
Degree of satisfaction from information (quality/quantity) provided about COVID-19 by the administrative office/managers of your hospital	5.00	4.79	2.91
Degree of satisfaction from the personal protective equipment against COVID-19 provided to you by your hospital	6.00	6.00	2.29
** *Degree of Experienced Distress* **			
Degree of emotional exhaustion	8.00	7.33	2.29
Degree of distress experienced from being avoided due to work in a COVID-19 healthcare setting during the pandemic	8.00	6.98	2.69

**Table 3 ijerph-19-11049-t003:** The mean, median, standard deviation, scale range and frequency of responses in each percentile of the total STSS-M score (Ν = 233).

			Scale	Range		Distribution of STSS-M Scores in Percentiles				
						Non-Clinical Symptoms		Clinical Symptoms		
	M (SD)	Median	Actual Scale Range	Observed Scale Range		25th	50th	75th	90th	95th
	45.38 (12.97)	46.00	17–85	17–80	**Total STSS-M score**	36	45	55	62	67
Frequency (%)							74.3%		25.7%	

M, mean; SD, standard deviation.

**Table 4 ijerph-19-11049-t004:** The correlation analysis between total STSS-M score and work-related satisfaction/distress variables (Pearson’s r; Ν = 233).

Work-Related Satisfaction/Distress Variables	Pearson’s r	*p*-Value
Degree of emotional exhaustion	0.490	<0.001
Degree of professional satisfaction	−0.298	<0.001
Degree of satisfaction from provided care in the last month	−0.201	0.002
Degree of satisfaction from information (quality/quantity) provided about COVID-19 by the administrative office/managers of your hospital	−0.204	0.002
Degree of satisfaction from the personal protective equipment against COVID-19 provided to you by your hospital	−0.232	<0.001
Degree of distress experienced from being avoided due to work in a COVID-19 healthcare setting during the pandemic	0.317	<0.001

**Table 5 ijerph-19-11049-t005:** The predictors of clinical trauma symptoms (N = 233).

	*p*-Value	Exp(B)	95% C.I. for EXP(B)	
			Lower	Upper
**Ranking** Head nurse/Under head nurse **Staff nurse**	0.025	7.669	1.285	45.755
**Emotional Exhaustion**	0.000	1.556	1.242	1.950
**Work Satisfaction**	0.013	0.817	0.696	0.959
**Distress Experienced from Being Avoided due to Work in a COVID-19 Wetting**	0.004	1.264	1.076	1.485
Constant	0.000	0.001		

## Data Availability

The datasets that were generated and/or analyzed during the present study are not publicly available because the authors are currently working on them in order to prepare additional publications; however, they are available from the corresponding author upon reasonable request.

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
