# Peer review of "Work-Related Traumatic Stress Response in Nurses Employed in COVID-19 Settings"

_ijerph, 2022, doi:10.3390/ijerph191711049_

Round 1
Reviewer 1 Report (Previous Reviewer 1)
The more careful management of data (numbers) needs. For instance, on lines 21-22, the authors mentioned that ¨two hundred and twenty-three nurses participated¨ in this study, but on line and all Tables the information referred to, number (233). In general, the use of statistics is bizzare. For instance, in the Table 1, the head "Frequency" of the second column does not correspond to the reported in numbers data of male/female participants. The simple visit to Google (Oxford Dictionary) shows the definition of frequency as the rate. In all Tables, the integer values should be reported on the left side of each mean values ( Benford´s law of digits scaling). Banni et al., (2021) paper, cited by the authors (42), is the best example of data managment.
The strong recommendation
Author Response
Comments by Reviewer # 1 |
|||
|
Reviewer’s Comment |
Response |
Citation in the text |
C1 |
The more careful management of data (numbers) needs. For instance, on lines 21-22, the authors mentioned that ¨two hundred and twenty-three nurses participated¨ in this study, but on line and all Tables the information referred to, number (233).
|
Thank you for this comment which helped us to revise any inconsistencies throughout the text. Specifically, “two hundred and twenty-three” was replaced by “two hundred and thirty-three”. Additional inconsistencies (minor numeric typos) were revised (please see underlined text). |
Page 1, Abstract, Line 21 (underlined text)
Page 1, Abstract, Line 28 Page 6, Results section, Lines 245-249, 262 Page 6, Table 1, Line 255 Page 7, Lines 268-270 Page 8, line 288 (underlined text)
|
C2 |
In general, the use of statistics is bizzare. For instance, in the Table 1, the head "Frequency" of the second column does not correspond to the reported in numbers data of male/female participants. The simple visit to Google (Oxford Dictionary) shows the definition of frequency as the rate. |
We thank the reviewer for this comment which helped us to increase the rigor of results presentation. Specifically, the term “Frequency” was replaced by “N” to accurately depict the reported in numbers data. Moreover, all Tables were checked accordingly. A relevant revision was applied in Table 3. Specifically “N” was eliminated, since the reported data are on rate (%). Moreover the 1st row was revised to reflected the presented data in more clear way. Specifically “Percentiles” was replaced by “Distribution of STSS-M score in percentiles”. The corresponding text was accordingly revised (page 7, lines 270-271) |
Page 6, Table 1, Line 255
Page 7, Table 3 (underlined text)
Page 7, lines 270-271 |
C3 |
In all Tables, the integer values should be reported on the left side of each mean values (Benford´s law of digits scaling). Banni et al., (2021) paper, cited by the authors (42), is the best example of data management.
|
Thank you for this comment which helped us to revise Table 2. Specifically, median values (integer values) have been moved to be reported on the left side of each mean values. |
Page 7, Table2 |

Reviewer 2 Report (Previous Reviewer 2)
Thank you for taking on board my earlier comments and suggestions, and those of other reviewers. I believe this is an interesting and well-written article, and one I hope the journal will see fit to publish.
Author Response
Comments by Reviewer # 2 |
|||
|
Reviewer’s Comment |
Response |
Citation in the text |
C2 |
Thank you for taking on board my earlier comments and suggestions, and those of other reviewers. I believe this is an interesting and well-written article, and one I hope the journal will see fit to publish. |
Thank you very much. |
|

Reviewer 3 Report (Previous Reviewer 3)
Congratulations, your manuscript has been significantly improved.
Author Response
Comments by Reviewer # 3 |
|||
|
Reviewer’s Comment |
Response |
Citation in the text |
C3 |
Congratulations, your manuscript has been significantly improved. |
Thank you very much. |
|

This manuscript is a resubmission of an earlier submission. The following is a list of the peer review reports and author responses from that submission.
Round 1
Reviewer 1 Report
There are two main problems with this contribution: 1. The first one is derived from the complex nature of traumatic stress responses in health care workers in COVID-19 setting; 2. The manuscript didn´t ANY original data used for statistical correlations and comparisons. The combined effects of individual (Table 1) and work-related factors (Tables 2-5) should be not only measured by corresponding correlation procedures, but also based on real data which can be viewed by each reader. The Tables 2-5 MUST include the first column with demographic and employment characteristics selected for this study and presented in Table 1. An example of correct presentation is the paper of Evanoff et al., 2020: "Work-related and personal factors associated with mental well-being during the COVID-19 response: survey of health care and other workers.", in: J. of Medical Internet Research, 22 (8): e21366. The so complex phenomena of SARS-CoV-2 require complex indexes for measuring or at least very carefully fullfilling the chosen statistical procedures. The simple Pearson coefficients comparison is insufficient for any decisive conclusion. This is mthe the possible reason for the vague conclusions of this work. The excelent paper of T.I.J. van der Berg et al., 2009: " The effects of work-related and individual factors on the Work Ability Index: a systematic review, Occup Environ Med is strongly recommended as an example to follow.
Author Response
On behalf of the co-authors, I would like to thank the reviewers for the hard work put on our manuscript. We tried our best to address all their comments in full extent in the provided time schedule. Please, find uploaded our revised manuscript entitled: “Work-related traumatic stress response in nurses employed in COVID-19 settings” and 3 separate files with our responses to each reviewer.

Reviewer 2 Report
In general, this was a well written article. I have made my comments as track changes on a PDF version of the article.

Author Response

(The authors gave the same response as above.)

Reviewer 3 Report
Dear colleagues, I hope this message find you well.
Thank you for giving me the opportunity of reading the work “Work-related traumatic stress response in nurses employed in COVID-19 settings”, it has been a very big pleasure to collaborate reviewing this manuscript. The topic of this paper is very interesting and it seems necessary to delve it. However, there are several questions to improve before to publish it. I would suggest some changes:
Title and abstract
· Ok
Introduction
· Dear colleagues, the structure of the introduction is not clear. I recommended to divide the introduction into several subsections. For example, creating a specific subsection where describe each group of variables/factors involved. Moreover, aims should be described in a clearly way.
· Secondly, when you explain the signs of psychological distress and mental health as a result of COVID-19, it is necessary to add more data and references. I recommend you to add this paper recently published (https://doi.org/10.3390/ijerph18147422), which proposes COVID-19 pandemic as a PTSD.
· Moreover, at lines page 2, introducing more studies could be interesting in order to support better how COVID-19 has psychologically affected health-care professionals:
https://doi.org/10.3390/jcm10184077
https://doi.org/10.1016/S2215-0366(20)30307-2
· Aims and hypotheses should be inside of introduction section.
Method
· To add some information about the sample in this section could be useful.
Results
· Have you assessed/controlled another variable in order to avoid interferences?
Discussion
· In my humble opinion, it could be useful to describe in more detail the practical and theoretical implications of this research. It would be useful they contextualize better the contribution within the framework of the issue explaining why the contribution is useful and enrich the impact.
Conclusions
· Nothing to add. Good job.
Author Response

(The authors gave the same response as above.)
